# SOFT CONVEX QUANTIZATION: REVISITING VECTOR QUANTIZATION WITH CONVEX OPTIMIZATION

## ABSTRACT

Vector Quantization (VQ) is a well-known technique in deep learning for extracting informative discrete latent representations. VQ-embedded models have shown impressive results in a range of applications including image and speech generation. VQ operates as a parametric K-means algorithm that quantizes inputs using a single codebook vector in the forward pass. While powerful, this technique faces practical challenges including codebook collapse, non-differentiability and lossy compression. To mitigate the aforementioned issues, we propose Soft Convex Quantization (SCQ) as a direct substitute for VQ. SCQ works like a differentiable convex optimization (DCO) layer: in the forward pass, we solve for the optimal convex combination of codebook vectors that quantize the inputs. In the backward pass, we leverage differentiability through the optimality conditions of the forward solution. We then introduce a scalable relaxation of the SCQ optimization and demonstrate its efficacy on the CIFAR-10, GTSRB and LSUN datasets. We train powerful SCQ autoencoder models that significantly outperform matched VQ-based architectures, observing an order of magnitude better image reconstruction and codebook usage with comparable quantization runtime.

## 1 INTRODUCTION

Over the past years, architectural innovations and computational advances have both contributed to the spectacular progress in deep generative modeling (Razavi et al., 2019; Esser et al., 2020; Rombach et al., 2021). Key applications driving this field include image (van den Oord et al., 2017; Esser et al., 2020; Rombach et al., 2021) and speech (Dhariwal et al., 2020) synthesis.

State-of-the-art generative models couple autoencoder models for compression with autoregressive (AR) or diffusion models for generation (van den Oord et al., 2017; Chen et al., 2017; Esser et al., 2020; Rombach et al., 2021; Gu et al., 2022). The autoencoder models are trained in the first stage of the generation pipeline and aim to extract compressed yet rich latent representations from the inputs. The AR or diffusion models are trained in the second stage using latents obtained from the pre-trained autoencoder and are used for generation. Throughout this work, we refer to the autoencoder models as *first-stage* models while the actual generative models trained on the latent space are referred to as *second-stage* models. Therefore, the effectiveness of the entire generation approach hinges upon the extraction of informative latent codes within the first stage. One of the pervasive means of extracting such latent codes is by embedding a vector quantization (VQ) bottleneck (van den Oord et al., 2017; Razavi et al., 2019) within the autoencoder models.

Motivated by the domain of lossy compression and techniques such as JPEG (Wallace, 1992), VQ is a method to characterize a discrete latent space. VQ operates as a parametric online K-means algorithm: it quantizes individual input features with the "closest" learned codebook vector. Prior to VQ, the latent space of variational autoencoders (VAEs) was continuous and regularized to approximate a normal distribution (Kingma et al., 2014; Kingma & Welling, 2019). The VQ method was introduced to learn a robust discrete latent space that doesn't suffer the posterior collapse drawbacks faced by VAEs regularized by Kullback-Leibler distance (van den Oord et al., 2017). Having a discrete latent space is also supported by the observation that many real-world objects are in fact discrete: images appear in categories and text is represented as a set of tokens. An additional benefit of VQ in the context of generation, is that it accommodates learning complex latent categorical priors. Due to these benefits, VQ underpins several image generation techniques including vector-quantized vari-

ational autoencoders (VQVAEs) (van den Oord et al., 2017; Razavi et al., 2019), vector-quantized generative adversarial networks (VQGANs) (Esser et al., 2020), and vector-quantized diffusion (VQ Diffusion) (Gu et al., 2022). Notably it also has application in text-to-image (Ramesh et al., 2021) and speech (Dhariwal et al., 2020) generation.

While VQ has been applied successfully across many generation tasks, there still exist shortcomings in the method. One practical issue pertains to backpropagation when learning the VQ model: the discretization step in VQ is non-differentiable. Currently, this is overcome by approximating the gradient with the straight-through estimator (STE) (Bengio et al., 2013). The VQ method is also plagued by the "codebook collapse" problem, where only a few codebook vectors get trained due to a "rich getting richer" phenomena (Kaiser et al., 2018). Here codebook vectors that lie closer to the distribution of encoder outputs get stronger training signals. This ultimately leads to only a few codebook vectors being used in the quantization process, which impairs the overall learning process. Another limitation with VQ is that inputs are quantized with exactly one (nearest) codebook vector (van den Oord et al., 2017). This process is inherently lossy and puts heavy burden on learning rich quantization codebooks. Several works have aimed at mitigating the aforementioned issues with heuristics (Jang et al., 2017; Maddison et al., 2017; Zeghidour et al., 2021; Dhariwal et al., 2020; Huh et al., 2023; Lee et al., 2022). While the recent works have demonstrated improvements over the original VQ implementation, they are unable to fully attain the desired behavior: exact backpropagation through a quantization step that leverages the full capacity of the codebook.

In view of the above shortcomings of existing VQ techniques, we propose a technique called soft convex quantization (SCQ). Rather than discretizing encoder embeddings with exactly one codebook vector, SCQ solves a convex optimization in the forward pass to represent each embedding as a convex combination of codebook vectors. Thus, any encoder embedding that lies within the convex hull of the quantization codebook is exactly representable. Inspired by the notion of differentiable convex optimization (DCO) (Amos & Kolter, 2017; Agrawal et al., 2019), this approach naturally lends itself to effectively backpropagate through the solution of SCQ with respect to the entire quantization codebook. By the means of this implicit differentiation, stronger training signal is conveyed to all codebook vectors: this has the effect of mitigating the codebook collapse issue.

We then introduce a scalable relaxation to SCQ amenable to practical codebook sizes and demonstrate its efficacy with extensive experiments: training 1) VQVAE-type models on CIFAR-10 (Krizhevsky, 2009) and German Traffic Sign Recognition Benchmark (GTSRB) (Stallkamp et al., 2012) datasets, and 2) VQGAN-type models (Esser et al., 2020) on higher-resolution LSUN (Yu et al., 2015) datasets. SCQ outperforms state-of-the-art VQ variants on numerous metrics with faster convergence. More specifically, SCQ obtains up to an order of magnitude improvement in image reconstruction and codebook usage compared to VQ-based models on the considered datasets while retaining a comparable quantization runtime. We also highlight SCQ's improved performance over VQ in low-resolution latent compression, which has the potential of easing the computation required for downstream latent generation.

## 2 BACKGROUND

### 2.1 VECTOR QUANTIZATION NETWORKS

Vector quantization (VQ) has risen to prominence with its use in generative modeling (van den Oord et al., 2017; Razavi et al., 2019; Rombach et al., 2021). At the core of the VQ layer is a codebook, i.e. a set of $K$ latent vectors $\mathcal{C} := \{c_j\}_{j=1}^K$ used for quantization. In the context of generative modeling, an encoder network $E_\phi(\cdot)$ with parameters $\phi$ maps input $x$ into a lower dimensional space to vector $z_e = E(x)$. VQ replaces $z_e$ with the closest (distance-wise) vector in the codebook:

$$z_q := Q\left(z_e\right) = c_k, \quad \text{where } k = \arg\min_{1 \le j \le K} \|z_e - c_j\|, \tag{1}$$

and $Q(\cdot)$ is the quantization function. The quantized vectors $z_q$ are then fed into a decoder network $D_\theta(\cdot)$ with parameters $\theta$, which aims to reconstruct the input $x$. Akin to standard training, the overarching model aims to minimize a task specific empirical risk:

$$\min_{\phi,\theta,\mathcal{C}} \mathbb{E}_{x \sim \mathcal{P}_{dist}}[\mathcal{L}_{\text{task}}(D(Q(E(x))), x)] \tag{2}$$

where $\mathcal{L}_{\text{task}}$ could be a reconstruction (van den Oord et al., 2017; Razavi et al., 2019) or perceptual loss (Esser et al., 2020; Rombach et al., 2021) and $\mathcal{P}_{dist}$ is the underlying data distribution. Training is performed using standard first-order methods via backpropagation (Rumelhart et al., 1986). As differentiation through the discretization step is ill-posed, the straight-through-estimator (STE) (Bengio et al., 2013) is used as a gradient approximation.

To ensure an accurate STE, a commitment loss is introduced to facilitate learning the codebook:

$$\mathcal{L}_{\text{commit}}(E_\phi, \mathcal{C}) = (1 - \beta)d(sg[z_e], z_q) + \beta d(z_e, sg[z_q]), \tag{3}$$

where $d(\cdot, \cdot)$ is a distance metric, $\beta > 0$ is a hyperparameter and $sg[\cdot]$ is the *stop gradient* operator. The first term brings codebook nearer to the encoder embeddings, while the second term optimizes over the encoder weights and aims to prevent fluctuations between the encoder outputs and its discretization. Combining $\mathcal{L}_{\text{task}}$ and $\mathcal{L}_{\text{commit}}$ yields a consolidated training optimization:

$$\min_{\phi, \theta, \mathcal{C}} \mathbb{E}_{x \sim \mathcal{P}_{dist}} \left[ \mathcal{L}_{\text{task}}(D_\theta(Q(E_\phi(x))), x) + \mathcal{L}_{\text{commit}}(E_\phi, \mathcal{C}) \right]. \tag{4}$$

A concrete example of this framework is the loss used to train the VQVAE architecture (van den Oord et al., 2017; Razavi et al., 2019):

$$\mathcal{L}_{\text{VQ}}(E_\phi, D_\theta, \mathcal{C}) = \|x - \hat{x}\|_2^2 + (1 - \beta)\|sg[z_e] - z_q\|_2^2 + \beta\|z_e - sg[z_q]\|_2^2. \tag{5}$$

where $\hat{x} := D_\theta(Q(E_\phi(x)))$ is the reconstruction.

## 2.2 VECTOR QUANTIZATION CHALLENGES

In the next subsections, we outline some of the main challenges faced by VQ, as well as relevant methods used to alleviate these.

### 2.2.1 GRADIENT APPROXIMATION

As mentioned in 2.1, differentiation through the discretization step is required to backpropagate through a VQ-embedded network. Taking the true gradient through the discretization would yield zero gradient signal and thus deter any useful model training potential. To this end, the STE is used to approximate the gradient. From the perspective of the discretization function, the upstream gradient is directly mapped to the downstream gradient during backpropagation, i.e. the non-differentiable discretization step is effectively treated as an identity map. While prior work has shown how a well-chosen coarse STE is positively correlated with the true gradient (Yin et al., 2019), further effort has been put into alleviating the non-differentiability issue. In Jang et al. (2017); Maddison et al. (2017), the Gumbel Softmax reparameterization method is introduced. This method reparameterizes a categorical distribution to facilitate efficient generation of samples from the underlying distribution. Let a categorical distribution over $K$ discrete values have associated probabilities $\pi_i$ for $i \in [K]$. Then we can sample from this distribution via the reparameterization:

$$\texttt{sample} \sim \arg\max_i \{G_i + \log \pi_i\}, \tag{6}$$

where $G_i \sim \text{Gumbel}(0, 1)$ are samples from the Gumbel distribution. Since the $\arg\max$ operator is not differentiable, the method approximates it with a Softmax operator during backpropagation.

### 2.2.2 CODEBOOK COLLAPSE

In the context of VQ, codebook collapse refers to the phenomenon where only a small fraction of codebook vectors are used in the quantization process (Kaiser et al., 2018). While the underlying cause is not fully understood, the intuition behind this behavior is that codebook vectors that lie nearer to the encoder embedding distribution receive more signal during training and thus get better updates. This causes an increasing divergence in distribution between embeddings and underused codebook vectors. This misalignment is referred to as an *internal codebook covariate shift* (Huh et al., 2023). Codebook collapse is an undesired artefact that impairs the overarching model's performance as the full codebook capacity is not used. Thus, there have been many concerted efforts to mitigate this issue. One line of work targets a codebook reset approach: replace the dead codebook vectors with a randomly sampled replacement vector (Zeghidour et al., 2021; Dhariwal et al.,

2020). This approach requires careful tuning of iterations before the replacement policy is executed. Another direction of work aims to maintain stochasticity in quantization during the training process (Kaiser et al., 2018; Takida et al., 2022). This body of work is based on observations that the quantization is stochastic at the beginning of training and gradually convergences to deterministic quantization (Takida et al., 2022). In (Huh et al., 2023), authors introduce an affine reparameterization of the codebook vectors to minimize the divergence of the unused codebook vectors and embedding distributions.

### 2.2.3 LOSSY QUANTIZATION

As mentioned in Section 2.2.1, VQ-embedded networks are trained with the STE that assume the underlying quantization function behaves like an identity map. Therefore, effective training relies on having a good quantization function that preserves as much information as possible of the encoder embeddings. Given an encoder embedding $z_e$, the quantized output can be represented as $z_q = z_e + \epsilon$ where $\epsilon$ is a measure of the residual error. Since STE assumes the quantization is an identity map, the underlying assumption is that $\epsilon = 0$. In practice, however, the quantization process with a finite codebook is inherently lossy and we have $\epsilon > 0$. Therefore, the underlying quantization function should make the quantization error as small as possible to guarantee loss minimization with the STE. For large residuals, no loss minimization guarantees can be made for the STE. Recent work has proposed an alternating optimization scheme that aims to reduce the quantization error for VQ (Huh et al., 2023). In (Lee et al., 2022), authors introduce residual quantization (RQ) which performs VQ at multiple depths to recursively reduce the quantization residual. While RQ has shown improved empirical performance, it is still plagued with the same core issues as VQ and trades-off additional computational demands for executing VQ multiple times within the same forward pass.

### 2.3 DIFFENTIABLE CONVEX OPTIMIZATION (DCO) LAYERS

DCO are an instantiation of implicit layers (Amos & Kolter, 2017) that enable the incorporation of constrained convex optimization within deep learning architectures. The notion of DCO layers was introduced in Amos & Kolter (2017) as quadratic progamming (QP) layers with the name OptNet. QP layers were formalized as

$$
\begin{aligned}
z_{k+1} := \arg\min_{z \in \mathbb{R}^n} \quad & z^\top R(z_k) z + z^\top r(z_k) \\
\text{s.t.} \quad & A(z_k) z + B(z_k) \leq 0, \\
& \bar{A}(z_k) z + \bar{B}(z_k) = 0
\end{aligned}
\tag{7}
$$

where $z \in \mathbb{R}^n$ is the optimization variable and layer output, while $R(z_k)$, $r(z_k)$, $A(z_k)$, $B(z_k)$, $\bar{A}(z_k)$, $\bar{B}(z_k)$ are optimizable and differentiable functions of the layer input $z_k$. Such layers can be naturally embedded within a deep learning architecture and the corresponding parameters can be learned using the standard end-to-end gradient-based training approach prevalent in practice. Differentiation with respect to the optimization parameters in equation 7 is achieved via implicit differentiation through the Karush-Kuhn-Tucker (KKT) optimality conditions (Amos & Kolter, 2017; Amos et al., 2017). On the computational side, Amos & Kolter (2017) develop custom interior-point batch solvers for OptNet layers that are able to leverage GPU compute efficiency.

## 3 METHODOLOGY

In this section, we introduce the soft convex quantization (SCQ) method. SCQ acts as an improved drop-in replacement for VQ that addresses many of the challenges introduced in Section 2.2.

### 3.1 SOFT CONVEX QUANTIZATION WITH DIFFERENTIABLE CONVEX OPTIMIZATION

SCQ leverages convex optimization to perform soft quantization. As mentioned previously, SCQ can be treated as a direct substitute for VQ and its variants. As such, we introduce SCQ as a bottleneck layer within an autoencoder architecture. The method is best described by decomposing its workings into two phases: the forward pass and the backward pass. The forward pass is summarized in Algorithm 1 and solves a convex optimization to perform soft quantization. The backward

---

**Algorithm 1** Soft Convex Quantization Algorithm

---

**Design choices:** Quantization regularization parameter $\lambda > 0$, embedding dimension $F$, codebook size $K$

**Input:** Encodings $Z_e \in \mathbb{R}^{N \times F \times \widetilde{H} \times \widetilde{W}}$

**Return:** Convex quantizations $Z_q \in \mathbb{R}^{N \times F \times \widetilde{H} \times \widetilde{W}}$

**Parameters:** Randomly initialize codebook $C \in \mathbb{R}^{D \times K}$

**begin forward**
1. $Z_e^{\text{flattened}} \in \mathbb{R}^{F \times N \widetilde{H} \widetilde{W}} \leftarrow \texttt{Reshape}(Z_e)$
2. $\widetilde{P} \in \mathbb{R}^{K \times N \widetilde{H} \widetilde{W}} \leftarrow \texttt{VQ}(\texttt{detach}(Z_e^{\text{flattened}}))$
3. $P^\star := \arg\min_{P \in \mathbb{R}^{K \times N \widetilde{H} \widetilde{W}}} \quad \|Z_e^{\text{flattened}} - CP\|_F^2 + \lambda\|P - \widetilde{P}\|_F^2 \ : \ P \geq 0, \ 1_K^\top P = 1_{N\widetilde{H}\widetilde{W}}$
4. $Z_q^{\text{flattened}} \in \mathbb{R}^{F \times N \widetilde{H} \widetilde{W}} \leftarrow CP^\star$
5. $Z_q \in \mathbb{R}^{N \times F \times \widetilde{H} \times \widetilde{W}} \leftarrow \texttt{Reshape}(Z_q^{\text{flattened}})$
**end**

---

pass leverages differentiability through the KKT optimality conditions to compute the gradient with respect to the quantization codebook.

**Forward pass.** Let $X \in \mathbb{R}^{N \times C \times H \times W}$ denote an input (e.g. of images) with spatial dimension $H \times W$, depth (e.g. number of channels) $C$ and batch size $N$. The encoder $E_\phi(\cdot)$ takes $X$ and returns $Z_e := E_\phi(X) \in \mathbb{R}^{N \times F \times \widetilde{H} \times \widetilde{W}}$ where $F$ is the embedding dimension and $\widetilde{H} \times \widetilde{W}$ is the latent resolution. $Z_e$ is the input to the SCQ. The SCQ method first runs VQ on $Z_e$ and stores the resulting one-hot encoding as $\widetilde{P} \in \mathbb{R}^{K \times N \widetilde{H} \widetilde{W}}$. $\widetilde{P}$ is detached from the computational graph and treated as a constant, i.e. no backpropagation through $\widetilde{P}$. Then $Z_e$ is passed into a DCO of the form

$$P^\star := \argmin_{P \in \mathbb{R}^{K \times N \widetilde{H} \widetilde{W}}} \quad \|Z_e^{\text{flattened}} - CP\|_F^2 + \lambda\|P - \widetilde{P}\|_F^2$$
$$s.t. \quad P \geq 0,$$
$$P^\top 1_K = 1_{N\widetilde{H}\widetilde{W}}, \tag{8}$$

where $Z_e^{\text{flattened}} \in \mathbb{R}^{F \times N \widetilde{H} \widetilde{W}}$ is a flattened representation of $Z_e$, $C \in \mathbb{R}^{F \times K}$ represents a randomly initialized codebook matrix of $K$ latent vectors, $\lambda > 0$ is a regularization parameter and $P \in \mathbb{R}^{K \times N \widetilde{H} \widetilde{W}}$ is a matrix we optimize over. SCQ solves convex optimization 8 in the forward pass: it aims to find weights $P^\star$ that best reconstruct the columns of $Z_e^{\text{flattened}}$ with a sparse convex combination of codebook vectors. Sparsity in the weights is induced by the regularization term that biases the SCQ solution towards the one-hot VQ solution, i.e. we observe that $\lim_{\lambda \to \infty} P^\star = \widetilde{P}$. The constraints in optimization 8 enforce that the columns of $P$ lie on the unit simplex, i.e. they contain convex weights. The codebook matrix $C$ is a parameter of the DCO and is updated with all model parameters to minimize the overarching training loss. It is randomly initialized before training and is treated as a constant during the forward pass. The SCQ output is given by $Z_q^{\text{flattened}} := CP^\star$. This is resolved to the original embedding shape and passed on to the decoder model.

**Backward pass.** During the forward pass SCQ runs VQ and then solves optimization 8 to find a sparse, soft convex quantization of $Z_e$. The underlying layer parameters $C$ are treated as constants during the forward pass. $C$ is updated with each backward pass during training. As $C$ is a parameter of a convex optimization, DCO enables backpropagation with respect to $C$ via implicit differentiation through the KKT conditions (Agrawal et al., 2019).

**Improved backpropagation and codebook coverage with SCQ.** During the forward pass of SCQ, multiple codebook vectors are used to perform soft quantization on $Z_e$. Optimization 8 selects a convex combination of codebook vectors for each embedding in $Z_e$. Therefore, SCQ is more inclined to better utilize the codebook capacity over VQ where individual codebook vectors are used for each embedding. Furthermore, owing to the DCO structure of SCQ, we can backpropagate effectively through this soft quantization step, i.e. training signal is simultaneously distributed across the entire codebook.

---

**Algorithm 2** Practical Soft Convex Quantization Algorithm

---

**Design choices:** Quantization regularization parameter $\lambda > 0$, number of projection steps $m$, embedding dimension $F$, codebook size $K$

**Input:** Encodings $Z_e \in \mathbb{R}^{N \times F \times \widetilde{H} \times \widetilde{W}}$

**Return:** Convex quantizations $Z_q \in \mathbb{R}^{N \times F \times \widetilde{H} \times \widetilde{W}}$

**Parameters:** Randomly initialize codebook $C \in \mathbb{R}^{F \times K}$

**begin forward**
1. $Z_e^{\text{flattened}} \in \mathbb{R}^{F \times N\widetilde{H}\widetilde{W}} \leftarrow \texttt{Reshape}(Z_e)$
2. $\widetilde{P} \in \mathbb{R}^{K \times N\tilde{H}\tilde{W}} \leftarrow \texttt{VQ}(Z_e^{\text{flattened}})$
3. $P \in \mathbb{R}^{K \times N\tilde{H}\tilde{W}} \leftarrow \texttt{LinearSystemSolver}(C^\top C + \lambda I, C^\top Z_e^{\text{flattened}} + \lambda\widetilde{P})$
**for** $i \in [m]$ **do**
    4. $P \leftarrow \max(0, P)$
    5. $P_{:,k} \leftarrow P_{:,k} - \frac{\sum_j P_{j,k}-1}{K}\mathbf{1}_K, \quad \forall k \in [N\widetilde{H}\widetilde{W}]$
**end for**
6. $P^\star \leftarrow P$
7. $Z_q^{\text{flattened}} \in \mathbb{R}^{F \times N\widetilde{H}\widetilde{W}} \leftarrow CP^\star$
8. $Z_q \in \mathbb{R}^{N \times F \times \widetilde{H} \times \widetilde{W}} \leftarrow \texttt{Reshape}(Z_q^{\text{flattened}})$
**end**

---

**Improved quantization with SCQ.** The goal of optimization 8 is to minimize the quantization error $\|Z_e - CP\|_F^2$ with convex weights in the columns of $P$. Thus, the optimization characterizes a convex hull over codebook vectors and can exactly reconstruct $Z_e$ embeddings that lie within it. This intuitively suggests SCQ's propensity for low quantization errors during the forward pass as compared to VQ variants that are inherently more lossy.

## 3.2 SCALABLE SOFT CONVEX QUANTIZATION

As proposed in (Amos & Kolter, 2017), optimization 8 can be solved using interior-point methods which give the gradients for free as a by-product. Existing software such as `CVXPYLayers` (Agrawal et al., 2019) is readily available to implement such optimizations. Solving 8 using such second-order methods incurs a cubic computational cost of $O((NK\widetilde{H}\widetilde{W})^3)$. However, for practical batch sizes of $N \approx 100$, codebook sizes $K \approx 100$ and latent resolutions $\widetilde{H} = \widetilde{W} \approx 50$, the cubic complexity of solving 8 is intractable.

To this end we propose a scalable relaxation of the optimization 8 that remains performant whilst becoming efficient. More specifically, we approximate 8 by decoupling the objective and constraints. We propose first solving the regularized least-squares objective with a linear system solver and then projecting the solution onto the unit simplex. With this approximation, the overall complexity decreases from $O((NK\widetilde{H}\widetilde{W})^3)$ for the DCO implementation to $O(K^3)$. In practice ($K \approx 10^3$) this linear solve adds negligible overhead to the wall-clock time as compared to standard VQ. This procedure is outlined in our revised scalable SCQ method shown in Algorithm 2. The projection onto the unit simplex is carried out by iterating between projecting onto the nonnegative orthant and the appropriate hyperplane.

## 4 EXPERIMENTS

This section examines the efficacy of SCQ by training autoencoder models in the context of generative modeling. Throughout this section we consider a variety of datasets including CIFAR-10 (Krizhevsky, 2009), the German Traffic Sign Recognition Benchmark (GTSRB) (Stallkamp et al., 2012) and higher-dimensional LSUN (Yu et al., 2015) Church and Classroom. We run all experiments on 48GB RTX 8000 GPUs.

## 4.1 TRAINING VQVAE-TYPE MODELS

We consider the task of training VQVAE-type autoencoder models with different quantization bottlenecks on CIFAR-10 (Krizhevsky, 2009) and GTSRB (Stallkamp et al., 2012). This autoencoder architecture is still used as a first stage within state-of-the-art image generation approaches such as VQ Diffusion (Gu et al., 2022). The autoencoder structure is depicted in Figure 4 in Appendix A and is trained with the standard VQ loss 5.

We compare the performance of SCQ against existing methods VQVAE (van den Oord et al., 2017), Gumbel-VQVAE (Jang et al., 2017), RQVAE (Lee et al., 2022), VQVAE with replacement (Zeghidour et al., 2021; Dhariwal et al., 2020), VQVAE with affine codebook transformation and alternating optimization (Huh et al., 2023). The autoencoder and quantization hyperparameters used for each dataset are detailed in Table 3 in Appendix A. The performance is measured using the reconstruction mean square error (MSE) and quantization error. The reconstruction error measures the discrepancy in reconstruction at the pixel level, while the quantization error measures the incurred MSE between the encoder outputs $Z_e$ and quantized counterpart $Z_q$. We also measure the perplexity of each method to capture the quantization codebook coverage. Larger perplexity indicates better utilization of the codebook capacity. In this experiment, the results on the test datasets were averaged over 5 independent training runs for 50 epochs. Table 1 presents the results.

Table 1: Comparison between methods on an image reconstruction task for CIFAR-10 and GTSRB over 5 independent training runs. The same base architecture is used for all methods. All metrics are computed and averaged on the test set.

| Method | CIFAR-10 | | | GTSRB | | |
|---|---|---|---|---|---|---|
| | MSE $(10^{-3})\downarrow$ | Quant Error$\downarrow$ | Perplexity$\uparrow$ | MSE $(10^{-3})\downarrow$ | Quant Error$\downarrow$ | Perplexity$\uparrow$ |
| VQVAE | 41.19 | 70.47 | 6.62 | 39.30 | 70.16 | 8.89 |
| VQVAE + Rep | 5.49 | $4.13 \times 10^{-3}$ | 106.07 | 3.91 | $1.61 \times 10^{-3}$ | 75.51 |
| VQVAE + Affine + OPT | 16.92 | $25.34 \times 10^{-3}$ | 8.65 | 11.49 | $13.27 \times 10^{-3}$ | 5.94 |
| VQVAE + Rep + Affine + OPT | 5.41 | $4.81 \times 10^{-3}$ | 106.62 | 4.01 | $1.71 \times 10^{-3}$ | 72.76 |
| Gumbel-VQVAE | 44.5 | $23.29 \times 10^{-3}$ | 10.86 | 56.99 | $47.53 \times 10^{-3}$ | 4.51 |
| RQVAE | 4.87 | $44.98 \times 10^{-3}$ | 20.68 | 4.96 | $38.29 \times 10^{-3}$ | 10.41 |
| SCQVAE | **1.53** | $\mathbf{0.15 \times 10^{-3}}$ | **124.11** | **3.21** | $\mathbf{0.24 \times 10^{-3}}$ | **120.55** |

SCQVAE outperforms all baseline methods across all metrics on both datasets. We observe in particular, the significantly improved quantization errors and perplexity measures. The improved quantization error suggests better information preservation in the quantization process, while improved perplexity indicates that SCQ enables more effective backpropagation that better utilizes the codebook's full capacity. These improvements were attained while maintaining training wall-clock time with the VQ baselines. The RQVAE method, on the other hand, did incur additional training time (approximately $2\times$) due to its invoking of multiple VQ calls within a single forward pass.

Figures 1 (a) and (b) illustrate SCQVAE's improved convergence properties over state-of-the-art RQVAE (Lee et al., 2022) and VQVAE with replacement, affine transformation and alternating optimization (Huh et al., 2023). For both datasets, SCQVAE is able to converge to a lower reconstruction MSE on the test dataset (averaged over 5 training runs).

We next considered the higher-dimensional ($256 \times 256$) LSUN (Yu et al., 2015) Church dataset. Again Table 3 summarizes the hyperparameters selected for the underlying autoencoder. Figure 2 visualizes the reconstruction of SCQVAE in comparison with VQVAE van den Oord et al. (2017) and Gumbel-VAE (Jang et al., 2017; Maddison et al., 2017) on a subset of test images after 1, 10 and 20 training epochs. This visualization corroborates previous findings and further showcases the rapid minimization of reconstruction MSE with SCQVAE. A similar visualization for CIFAR-10 is given in Figure 5 in Appendix A.

**SCQ Analysis.** To better understand how SCQ improves codebook perplexity, Figure 3 visualizes image reconstruction of SCQVAE for a varying number of codebook vectors. The SCQVAE is trained on LSUN Church according to Table 3. More specifically, we restrict the number of codebook vectors used for soft quantization to the top-$S$ most significant ones with $S \in [25]$. Here significance is measured by the size of the associated weight in matrix $P^\star$ of Algorithm 2. We observe a behavior analogous to principal components analysis (PCA): the reconstruction error re-

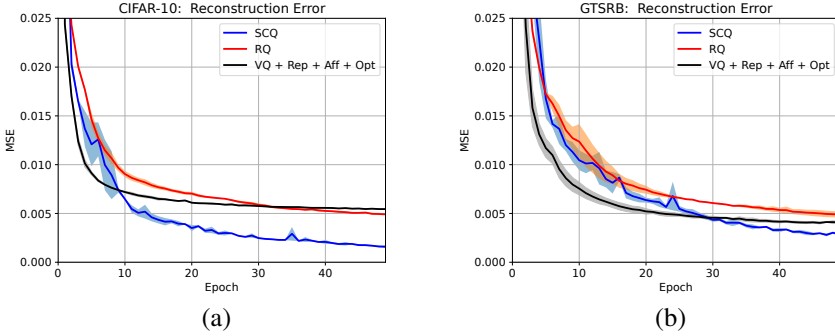

Figure 1: SCQVAE's improved convergence of test reconstruction MSE on CIFAR-10 (a) and GT-SRB (b).

duces as more codebook vectors are used in the quantization process. This suggests the quantization burden is carried by multiple codebook vectors and explains the improved perplexity of SCQ seen in Table 1. The number of codebook vectors needed for effective reconstruction depends on the sparsity induced in SCQ via hyperparameter $\lambda$.

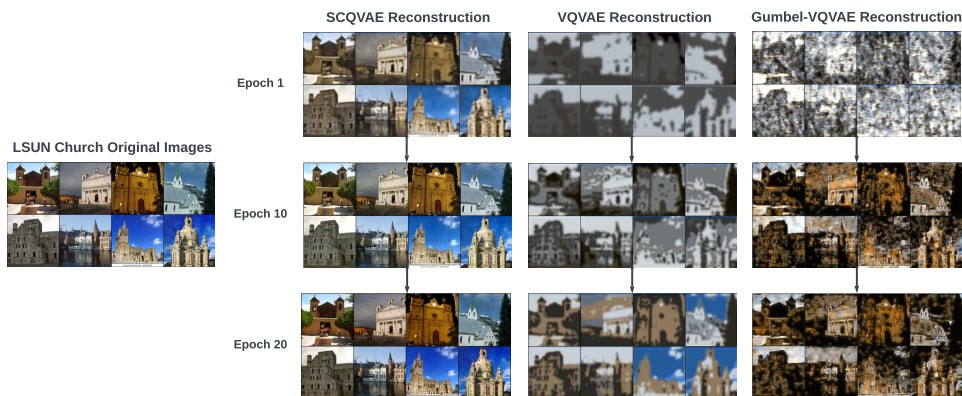

Figure 2: Comparison of LSUN (Yu et al., 2015) Church reconstruction on the test dataset.

## 4.2 TRAINING VQGAN-TYPE MODELS

In this section, we focus on training first-stage models used within image synthesis methods such as unconditional latent diffusion models (LDM). We use the LSUN Yu et al. (2015) Church and Classroom datasets and train VQGAN-type architectures trained with the associated VQGAN loss Esser et al. (2020). When using the SCQ quantizer, we refer to the aforementioned architectures as SCQGAN models.

To truncate training time, we use $5 \times 10^4$ randomly drawn samples from the LSUN Church and Classroom datasets to train VQGAN (Esser et al., 2020) and SCQGAN models. Table 4 summarizes the hyperparameter configuration used for both the VQ and SCQGAN architectures. The performance of both archictures was measured on the test set with the VQGAN loss (Esser et al., 2020) referred to as $\mathcal{L}_{\mathrm{VQGAN}}$, and LPIPS (Zhang et al., 2018). To examine the efficacy of both methods at different latent compression resolutions we train different architectures that compress the $256 \times 256$ images to $64 \times 64$, $32 \times 32$ and $16 \times 16$ dimensional latent resolutions. Table 2 summarizes the results. SCQGAN outperforms VQGAN on both datasets across both metrics on all resolutions. This result highlights the efficacy of SCQ over VQ in preserving information during quantization - especially at smaller resolution latent spaces (greater compression). This result is particularly exciting for downstream tasks such as generation that leverage latent representations. More effective

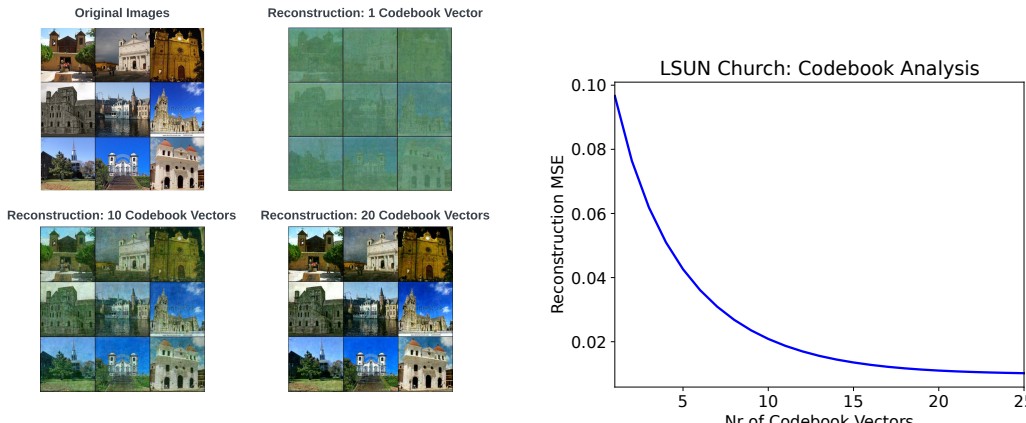

Figure 3: SCQ reconstruction performance with a varying number of codebook vectors on the LSUN Church dataset.

compression potentially eases the computational burden on the downstream tasks whilst maintaining performance levels. Figures 6-8 in Appendix B further illustrate faster convergence for SCQGAN on both metrics across all resolutions.

Table 2: Comparison between SCQGAN and VQGAN on LSUN image reconstruction tasks. The same base architecture is used for all methods and metrics are computed on the test set.

| Method | LSUN Church | | LSUN Classroom | |
|--------|-------------|--|----------------|--|
| | $\mathcal{L}_{\text{VQGAN}}$ $(10^{-1})\downarrow$ | LPIPS $(10^{-1})\downarrow$ | $\mathcal{L}_{\text{VQGAN}}$ $(10^{-1})\downarrow$ | LPIPS $(10^{-1})\downarrow$ |
| VQGAN (64-d Latents) | 4.76 | 4.05 | 3.50 | 3.28 |
| SCQGAN (64-d Latents) | **3.93** | **3.88** | **3.29** | **3.23** |
| VQGAN (32-d Latents) | 6.60 | 6.22 | 8.01 | 7.62 |
| SCQGAN (32-d Latents) | **5.53** | **5.48** | **6.76** | **6.53** |
| VQGAN (16-d Latents) | 8.32 | 8.18 | 9.87 | 9.68 |
| SCQGAN (16-d Latents) | **7.86** | **7.84** | **9.19** | **9.15** |

## 5 CONCLUSION

This work proposes soft convex quantization (SCQ): a novel soft quantization method that can be used as a direct substitute for vector quantization (VQ). SCQ is introduced as a differentiable convex optimization (DCO) layer that quantizes inputs with a convex combination of codebook vectors. SCQ is formulated as a DCO and naturally inherits differentiability with respect to the entire quantization codebook. This enables overcoming issues such as inexact backpropagation and codebook collapse that plague the VQ method. Moreover, SCQ is able to exactly represent inputs that lie within the convex hull of the codebook vectors, which mitigates lossy compression. Experimentally, we demonstrate that a scalable relaxation of SCQ facilitates improved learning of autoencoder models as compared to baseline VQ variants on CIFAR-10, GTSRB and LSUN datasets. SCQ gives up to an order of magnitude improvement in image reconstruction and codebook usage compared to VQ-based models on the aforementioned datasets while retaining comparable quantization runtime.

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

## A  FURTHER DETAILS ON VQVAE EXPERIMENTS

In Figure 4 we illustrate the autoencoder architecture used in experiments described in Section 4.1. The hyperparameters described in van den Oord et al. (2017) were adjusted for the considered datasets. Table 3 describes the hyperparameters used in the experiment. Figure 5 visualizes the reconstruction of CIFAR-10 (Krizhevsky, 2009) test images by the SCQVAE, VQVAE (van den Oord et al., 2017) and Gumbel-VQVAE (Jang et al., 2017; Maddison et al., 2017) architectures. The visualization shows significantly improved reconstruction performance of SCQVAE over the baselines.

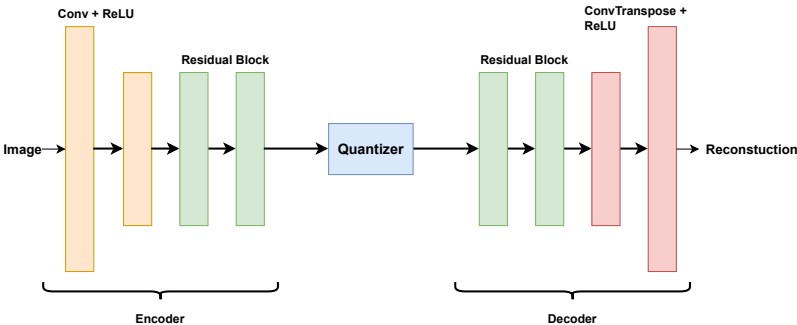

Figure 4: Autoencoder architecture for the reconstruction experiments on the CIFAR-10 (Krizhevsky, 2009) and GTSRB (Stallkamp et al., 2012) datasets.

Table 3: Hyperparameters of autoencoder used for CIFAR-10 (Krizhevsky, 2009), GTSRB (Stallkamp et al., 2012) and LSUN (Yu et al., 2015) Church experiments. $\lambda$ and $m$ are only applicable to the SCQ architecture.

|  | CIFAR-10 | GTSRB | LSUN Church |
|---|---|---|---|
| Image size | $32 \times 32$ | $48 \times 48$ | $256 \times 256$ |
| Latent size | $16 \times 16$ | $24 \times 24$ | $64 \times 64$ |
| $\beta$ (def. in equation 5) | 0.25 | 0.25 | 0.25 |
| Batch size | 128 | 128 | 128 |
| Conv channels | 32 | 32 | 128 |
| Residual channels | 16 | 16 | 64 |
| Nr of residual blocks | 2 | 2 | 2 |
| Codebook size | 128 | 128 | 128 |
| Codebook dimension | 16 | 16 | 32 |
| $\lambda$ (in Algorithm 2) | 0.1 | 0.1 | 0.1 |
| $m$ (in Algorithm 2) | 20 | 20 | 20 |
| Training steps | 19550 | 10450 | 7850 |

## B   FURTHER DETAILS ON VQGAN EXPERIMENTS

Table 4: Hyperparameters of VQ/SCQGAN models trained on the LSUN (Yu et al., 2015) Church and Classroom datasets. Images were center-cropped to a size $256 \times 256$. Models were trained to compress latents to different resolutions. $\lambda$ and $m$ are only applicable to the SCQ architecture.

|  | $16 \times 16$ Latents | $32 \times 32$ Latents | $64 \times 64$ Latents |
|---|---|---|---|
| $\beta$ (def. in equation 5) | 0.25 | 0.25 | 0.25 |
| Batch size | 64 | 64 | 64 |
| Base residual channels (C) | 128 | 128 | 128 |
| Residual channels at different resolutions | [C, 2C, 4C, 8C, 8C] | [C, 2C, 4C, 8C] | [C, 2C, 4C] |
| Nr of residual blocks | 2 | 2 | 2 |
| Codebook size | 512 | 512 | 1024 |
| Codebook dimension | 10 | 8 | 3 |
| $\lambda$ (in Algorithm 2) | 0.1 | 0.1 | 0.1 |
| $m$ (in Algorithm 2) | 2 | 2 | 2 |
| Total Params ($10^6$) | 339 | 225 | 58 |
| Training steps | 9384 | 9384 | 9384 |

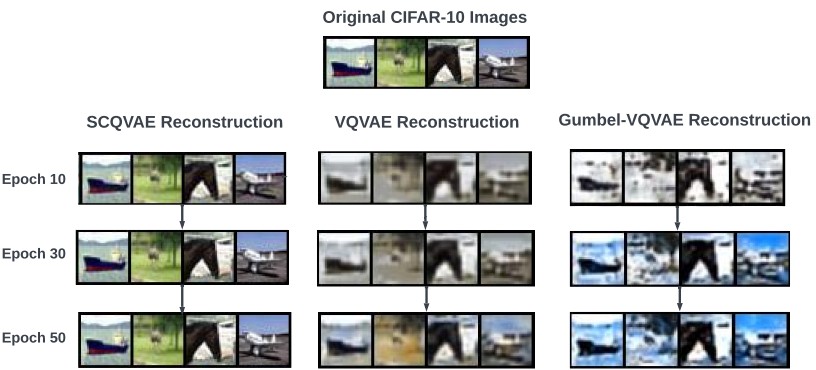

Figure 5: Comparison of CIFAR-10 (Krizhevsky, 2009) reconstruction on the validation dataset.

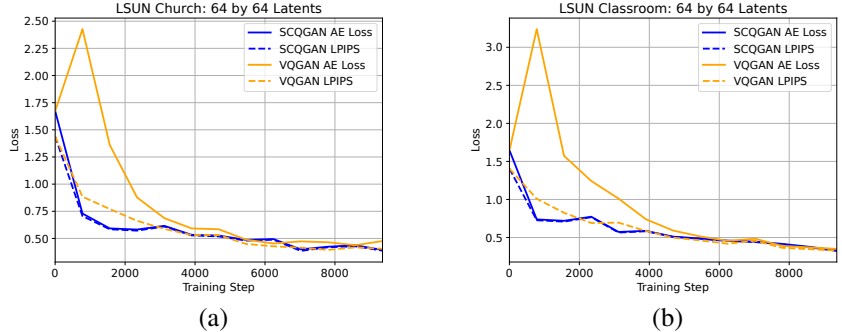

Figure 6: SCQGAN outperforms VQGAN on $\mathcal{L}_{\text{VQGAN}}$ (AE loss) and LPIPs on the LSUN Church (a) and Classroom (b) datasets. These results are for a latent resolution of $64 \times 64$.

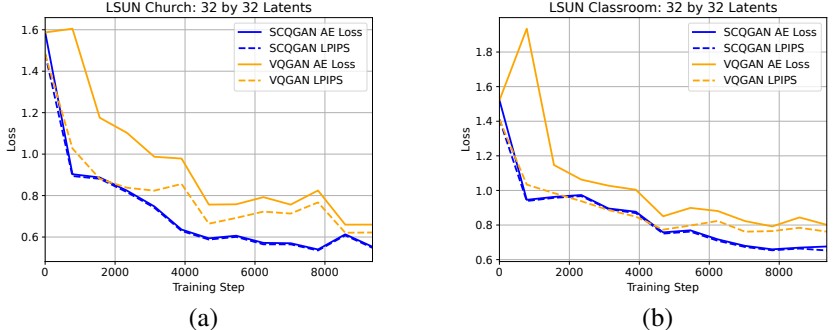

Figure 7: SCQGAN outperforms VQGAN on $\mathcal{L}_{\text{VQGAN}}$ (AE loss) and LPIPs on the LSUN Church (a) and Classroom (b) datasets. These results are for a latent resolution of $32 \times 32$.

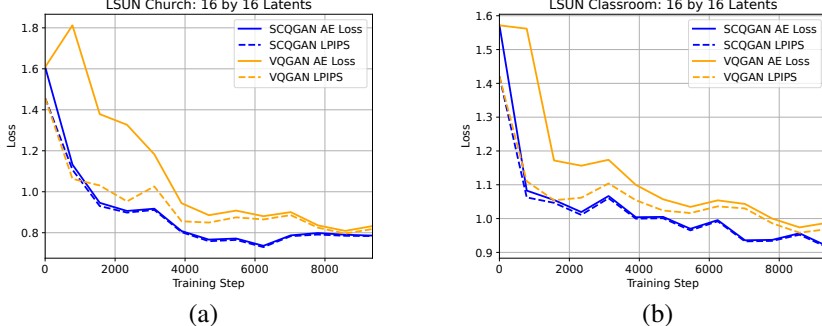

(a)                                                          (b)

Figure 8: SCQGAN outperforms VQGAN on $\mathcal{L}_{\text{VQGAN}}$ (AE loss) and LPIPs on the LSUN Church (a) and Classroom (b) datasets. These results are for a latent resolution of $16 \times 16$.

