# OpenReview forum: "Soft Convex Quantization: Revisiting Vector Quantization with Convex Optimization"
_ICLR.cc/2024/Conference — ICLR 2024 Conference Withdrawn Submission_

### Official Review · Reviewer_RwMk · 2023-10-30

**Soundness:** 3 good
**Presentation:** 3 good
**Contribution:** 3 good
**Rating:** 6
**Confidence:** 4

**Summary:**

This paper proposes a Soft Convex Quantization (SCQ) scheme to substitute the vector quantization (VQ) module in generative modeling. The authors first outline the challenges in VQ, including gradient approximation, codebook collapse, and lossy quantization, as well as recent related works for alleviating these issues. The proposed SCQ leverages convex optimization to perform soft quantization, which acts as an improved drop-in replacement for VQ that addresses many of the challenges. Experimental results demonstrate that SCQ is superior to existing VQ methods on various datasets.

**Strengths:**

1. The paper is well written and easy to follow.
2. The proposed SCQ is a novel method to mitigate the challenges in VQ mentioned above. First, the optimization of SCQ (via Eq. (8)) selects a convex combination of codebook vectors for each embedding, which means that the codebook capacity can be sufficiently utilized to avoid codebook collapse. Second, SCQ is implemented via Differentiable Convex Optimization (DCO), which enables effective and efficient backpropagate through soft quantization.
3. Extensive experiments are conducted to validate the effectiveness of the proposed SCQ. SCQ outperforms recent state-of-the-art methods on two different datasets in image reconstruction task in terms of three measurements. Analysis of SCQ presents that SCQ improves codebook perplexity and convergence during training. Besides, when combined with the training first-stage models, the authors show that SCQGAN performs better than VQGAN.

**Weaknesses:**

1. Soft quantization is also used to solve the gradient approximation challenge discussed in section 2.2.1, which is not mentioned in the paper.
2. The authors claim that SCQ is implemented with DCO. However, the relationship between the optimization of SCQ and DCO is ambiguous, especially the relationship between Eq. (7) and Eq. (8).
3. The analysis of how SCQ addresses the lossy quantization challenge compared with existing VQ methods is not convincing. More theoretical analysis or experiments are encouraged to demonstrate this point.
4. Some details are missing. For example, in Eq.(8), the codebook used to obtain \tilde(P) is not specified. Besides, in section 4.2, the training loss of SCQGAN is not given.

**Questions:**

1. What is the relationship between the optimization of SCQ and DCO?
2. Could the authors provide more detailed analysis on how SCQ mitigates the lossy quantization challenge?
3. Could the author give the missing details mentioned in weaknesses #4?

---

> ### Author Response · Authors · 2023-11-15
>
> We thank the reviewer for their comments and feedback. We address the reviewer’s concerns and questions below.
>
> **Regarding weaknesses:**
> 1. We would appreciate it if the reviewer could provide more clarity on which soft quantization method is being referred to.
> 2. The SCQ is formulated as an instantiation of a DCO layer. Specifically, in Eq. (7) we introduce the general notation for a DCO layer as proposed in (Amos & Kolter, 2017). In Eq. (8) we introduce the SCQ instantiation. In the forward pass we solve for the matrix $P$ containing convex combination weights in its columns and in the backward pass we update matrix $C$ which represents the parameters of the DCO layer. We then proceed to introduce a scalable version of the SCQ algorithm which relaxes the optimization given in Eq. (8). This scalable algorithm eliminates the need to solve a quadratic program in the forward pass to carry out the soft quantization. The scalable algorithm is presented in Alg. 2.
> 3. SCQ sees an improvement in the quantization performance over VQ. To make this clearer, we first consider how the quantization error is incurred for VQ: the error is measured between the input feature and the “closest” codebook vector. For SCQ, any individual input feature can be exactly reconstructed if it lies within the convex hull of the set of codebook vectors. This is a consequence of how the SCQ optimization in Eq. (8) is formulated. Thus, for any input feature that does not coincide exactly with a codebook vector and lies within the convex hull of codebook vectors, we incur no quantization error in SCQ whereas we incur nonzero error in the VQ method. For input features outside the convex hull of the set of codebook vectors, we again incur smaller error for SCQ as we measure the error with respect to the projection onto the convex hull. We would be happy to formalize this analysis and include it in the paper. We also demonstrate the improvement in quantization error empirically: in Table 1 (pg. 7, reproduced in General Comments section above), we have included the quantization error as a metric to evaluate SCQ’s quantization performance over competing VQ variants. The quantization error measures the incurred MSE between the encoder outputs and quantized counterparts. The table summarizes results on the image reconstruction performance on CIFAR-10 and GTSRB datasets. Across both datasets SCQ significantly outperforms competing methods on the induced quantization error.
> 4. We thank the reviewer for bringing this to our attention. The codebook used to obtain $\tilde{P}$ is the same codebook used throughout the SCQ process, i.e. codebook $C$. $\tilde{P}$ represents the output of the VQ method given the same codebook. We regularize the SCQ optimization in Eq. (8) with $\tilde{P}$ to bias the SCQ solution towards a one-hot encoding (this is to ensure compatibility with downstream autoregressive generation tasks). The loss curves of SCQGAN are given in Figures 6-8 in Appendix B of the paper.
>
>
> **Regarding questions:**
>
> 1. Please see the response to (2) above.
> 2. Please see the response to (3) above.
> 3. Please see the response to (4) above.

---

### Official Review · Reviewer_7Ro3 · 2023-10-31

**Soundness:** 3 good
**Presentation:** 2 fair
**Contribution:** 3 good
**Rating:** 6
**Confidence:** 4

**Summary:**

This work presents a differentiable and convex optimization layer as a direct substitute for vector quantization (VQ). It addresses three major challenges in classical VQ: a) non-differentiable k-means is replaced by softmax and gumbel sampling; b) codebook collapse through stochasticity; c) transforms the NP-hard quantization centroid generation into a convex hull over codebook vectors. The results outperforms other VQ-based architectures for image processing tasks.

**Strengths:**

1. This work demonstrated a great alternative to k-means based VQ training which is differentiable and convex.

2. Clean formulation on the optimization goals and algorithms.

3. Clear performance wins over other VQ techniques for neural networks.

**Weaknesses:**

1. In each comparison table, the non-quantized baseline numbers are missing.

2. Author emphasizes the problem of codebook collapse, but there were no quantitative support for how SCQ performs better than traditional k-means or VQ.

**Questions:**

1. Please elaborate on how SCQ prevents the codebook collapse problem.

2. What is the runtime latency of this work when compared to the uncompressed baseline?

---

> ### Author Response · Authors · 2023-11-15
>
> We thank the reviewer for their comments and feedback. We address the reviewer’s concerns and questions below.
>
> **Regarding weaknesses:**
> 1. For the CIFAR-10 and GTSRB experiments (see Section 4.1), we provide the non-quantized baseline numbers below. The same base autoencoder (AE) was used without any quantization bottleneck. Thus, we report only the final test MSE over both datasets as the other metrics are not applicable (i.e. there is no quantization or codebook). We would like to highlight that in Table 1 (pg. 7) we demonstrate that SCQ comes closest to this non-quantized MSE. Moreover, we would also like to emphasize that this baseline non-quantized autoencoder is not compatible with downstream generative applications due to the missing latent structure provided by a quantization bottleneck.
>
> **Results on CIFAR-10**
> | Method      | MSE ($10^{-3}$)$\downarrow$ | Quant Error$\downarrow$ | Perplexity$\uparrow$ | Avg Quant Time (ms)$\downarrow$ |
> | ----------- | ----------- | ----------- | ----------- | ----------- |
> | Non-quantized AE  | 0.44 | - | - | - |
>
> **Results on GTSRB**
> | Method      | MSE ($10^{-3}$)$\downarrow$ | Quant Error$\downarrow$ | Perplexity$\uparrow$ | Avg Quant Time (ms)$\downarrow$ |
> | ----------- | ----------- | ----------- | ----------- | ----------- |
> | Non-quantized AE  | 1.25 | - | - | - |
>
> 2. In Table 1 (pg. 7, reproduced in General Comments section above) we include the perplexity metric to quantitatively measure SCQ’s improvement over VQ and other variants with regards to the codebook collapse problem. The table compares methods on an image reconstruction task for CIFAR-10 and GTSRB and reports the average performance over 5 independent training runs. In the context of vector quantization, the perplexity metric measures the average usage of each codebook vector over a batch of samples: higher perplexity scores indicate better coverage of the entire codebook in the quantization process. Across both datasets SCQ significantly outperforms competing methods on the perplexity score.
>
> **Regarding questions:**
> 1. For every individual input feature, VQ utilizes the single “closest” codebook vector in the quantization. Thus, during training, the subset of codebook vectors that have been selected for a batch of inputs receive training signals and are updated. The codebook vectors that have not been selected do not receive any update signal. The codebook collapse problem is a consequence of this phenomenon, where only a small fraction of codebook vectors are repeatedly “chosen” and updated by VQ whereas the remainder are unused. SCQ overcomes this issue by representing each individual input feature with a (convex) combination of the entire codebook. Thus, during backpropagation, the entire codebook will receive a signal to update. We appreciate the reviewer’s question and hope this provides sufficient clarification. We also strive to make this more clear within the paper.
> 2. Please see the general comments for this question. We have extended Table 1 (pg. 7) with latency measures of the quantization methods.

---

> > ### Comment · Reviewer_7Ro3 · 2023-11-20
> >
> > I thank the authors for their rebuttal. However, for the first question on the "codebook collapse", please support your arguments with a quantitative study. Explanation without numbers is a bit thin for your claim.
> >
> > I also share same concern with reviewer Jz15 and d7nB. Authors should choose one of the two options in presenting this work:
> > a) Demonstrate quality under the same bit-rate (current SCQ is using way more bits than VQ), or
> > b) focus a particular downstream application that is only feasible/tractable using SCQ.

---

### Official Review · Reviewer_d7nB · 2023-10-31

**Soundness:** 2 fair
**Presentation:** 2 fair
**Contribution:** 1 poor
**Rating:** 3
**Confidence:** 3

**Summary:**

The authors propose soft convex quantization (SCQ) as a drop-in replacement for vector quantization (VQ) for generative modeling with architectures such as VQVAEs and VQGANs.

SCQ learns a codebook of vectors similar to VQ. However, SCQ uses its code vectors to define a convex polytope. Then, to "quantize" an arbitrary vector, SCQ projects it onto the closest point of this convex polytope. In particular, SCQ "quantizes" vectors inside the polytope with essentially zero error. This approach starkly contrasts with VQ, which quantizes an arbitrary vector to its closest code vector.

While SCQ's projection operation is differentiable (as opposed to VQ, which needs to use a straight-through estimator to perform gradient descent-based optimization), the authors note that it is not scalable. Thus, they develop a scalable approximation to it.

The authors perform image reconstruction experiments on several datasets and show that their method compares favorably to VQ-based methods.

**Strengths:**

I found using the code vectors to define a convex polytope and "quantize" vectors by projecting them onto the polytope quite interesting.

Furthermore, the authors compare their methods across several datasets using several different generative models.

**Weaknesses:**

I also find soft convex quantization a misnomer because the method doesn't quantize vectors (i.e., it doesn't map them to a discrete set of representations); it projects them onto a polytope. This latter issue is especially problematic because it means that comparing SCQ to VQ is meaningless.

From the perspective of autoencoders, SCQ is a worse model because it has zero error inside its convex polytope, but it introduces a projection error outside of it. Furthermore, it is computationally more expensive to use SCQ than just using a standard autoencoder.

However, theoretical issues aside, my primary problem with the paper is that the model's use cases are unclear.

As I understand, VQ is usually used for generative modeling or data compression. Since SCQ produces continuous "quantized" representations, it cannot be used for compression, so its use case seems to be restricted to generative modeling. However, the authors do not perform any experiments on generative modeling, only image reconstruction, so SCQ's performance for this task is unclear.

Could the authors please correct me if I misunderstood their approach? If not, could they please clarify what use case they intend for their method?

Besides this, the writing needs to be improved. The authors use non-standard terminology, e.g. they use the terms "first-stage" and "second-stage" models for inference and generative networks, respectively. There are many strangely worded sentences that are difficult to understand.

They also seem to misuse the term "sparsity" in section 3.1. A sparse vector has most of its entries be exactly zero, while the authors seem to mean close to zero.

In the same paragraph, the symbol C is overloaded to mean the number of input channels and the codebook matrix.

Finally, the authors don't provide any formal analysis for or perform any ablation studies between the "proper" convex optimization procedure in Algorithm 1 and the relaxed version they propose in Algorithm 2, so the error introduced by this approximation is unclear.

**Questions:**

n/a

---

> ### Author Response · Authors · 2023-11-15
>
> We thank the reviewer for their comments and feedback. We address the reviewer’s concerns below.
>
> **SCQ Motivation**
>
> We appreciate the reviewer’s concern regarding the name of our proposed algorithm. We believe that the best way of justifying the chosen name is by motivating it again. We propose SCQ as a drop-in replacement that relaxes VQ. VQ operates as a parametric online K-means algorithm: it quantizes individual input features with the single ”closest” learned codebook vector. However, due to challenges that arise from discretization with a single codebook vector (e.g. codebook collapse, inexact backpropagation, lossy quantization) our aim was to relax the hard quantization constraint of VQ to a “soft” one that involves multiple codebook vectors simultaneously. The “convex” aspect to our algorithm was born because we wanted probabilities as weights when combining codebook vectors to represent input features. This is a relaxation of VQ which is restricted to using one-hot encodings. We decided on using “convex” weights/probabilities as this would make the latent space compatible with downstream autoregressive generative models. We understand the reviewer’s concern that SCQ does not perform hard quantization. However, we maintain that our algorithm is indeed a relaxation/softening of the hard VQ, where we can trivially recover the VQ solution through SCQ by taking hyperparameter $\lambda$ in Eq. (8) to infinity. Thus, we believe that within this context it is also fair to view VQ as an instantiation of SCQ and directly compare the methods.
>
> **SCQ Application**
>
> As mentioned above, we intend for SCQ to be used as a drop-in replacement for VQ in the generative modeling framework. We regard this paper as the first in a sequence of works that explores the potential of the SCQ algorithm in this context. Specifically, in this paper our aim was to introduce SCQ as a relaxation of VQ that addresses many of the latter’s drawbacks. The goal was not to focus on a downstream generation application, but rather demonstrate SCQ’s potential as a soft quantization layer that enables training superior VAE architectures on several vision datasets whilst remaining compatible (by design) with downstream generative models. SCQ is readily compatible with latent diffusion processes that operate directly on the encoder embeddings, as well as autoregressive models that require supervision from a categorical distribution over the latent space. On the other hand, uncompressed baseline autoencoders are not compatible with downstream autoregressive models as there is no characterization of a categorical distribution over the latent space. We believe that the far superior performance of SCQ-embedded VAEs will lend itself naturally to improved results for downstream generation applications that utilize these models. In particular, we demonstrate in Section 4.2 how SCQGANs retain higher performance over VQGANs when considering smaller latent resolutions. This observation is particularly useful for generation tasks, as operating on smaller resolutions significantly reduces the required downstream computation. In future work, we aim to couple SCQ-embedded models with latent generative processes and showcase improved performance of SCQ on downstream applications.
>
> **Other**
>
> We thank the reviewer for pointing out some unclear/non-standard terminology, and overloaded notation. We aim to correct the writing accordingly.

---

> > ### Comment · Reviewer_d7nB · 2023-11-18
> > **Response to the authors**
> >
> > I thank the authors for their rebuttal. However, sadly, it did not change my view of the paper in its current form. I understand that SCQ is a relaxation of quantization but a very significant relaxation. It changes the range of the "quantization" layer from discrete (in the case of VQ) to continuous and allows for cases that couldn't remotely occur for VQ, such as having zero quantization error for every point inside the convex hull of the codebook vectors. This fact makes the comparison between VQ and SCQ unfair.
> >
> > As such, it seems like SCQ gets the worst of both worlds: 1) it is not as powerful as a simple autoencoder since it introduces an error for representations outside the convex hull of the codebook vectors, yet it is more challenging to train. 2) On the other hand, it uses continuous latent representations, so it cannot be used for compression.
> >
> > Therefore, I cannot see any situation in which I would choose to use SCQ over either VQ or simply not quantizing the latents. Hence, the authors should demonstrate what advantageous theoretical/empirical properties SCQ has (e.g., what is gained by restricting the decoder's domain to be compact / to be a polytope? What do the codebook vectors decode to compared to VQ?) or rewrite the paper to focus fully around a particular downstream application that is only feasible/tractable using SCQ.

---

### Official Review · Reviewer_Jz15 · 2023-11-01

**Soundness:** 3 good
**Presentation:** 3 good
**Contribution:** 3 good
**Rating:** 3
**Confidence:** 3

**Summary:**

This work proposes a continuous relaxation of vector quantization, where rather than assigning a vector to a single codebook element, the vector can be assigned to anywhere within the convex hull of codebook elements. This makes the resulting VQ variant easier to integrate into SGD-based training routines, increases codebook utilization due to the use of soft assignments, and increases the quality of the quantized representation.

**Strengths:**

1. Strong experimental results and useful illustrations for image reconstruction
1. Strong mathematical exposition

**Weaknesses:**

1. The comparison against VQ-based quantizers for image reconstruction quality needs to be justified more. Using VQ on a K-vector codebook leads to $\log K$ bits per codeword, while this technique seems to use $O(K)$ bits, due to its soft assignments (unless the assignments are rounded later on, which I don't believe is the case). It therefore seems rather obvious and trivial that this much higher-bitrate representation leads to better image reconstruction; it simply isn't much of an information bottleneck, relative to traditional VQ. Either:

    * Justify the comparison on image reconstruction quality, when the baseline algorithms seem to be at a severe disadvantage given their much lower bitrate.
    * Show superior results on downstream tasks such as image generation, where compressed representation isn't the goal but rather a means to achieve the desired result.
1. No mention of training time. The scalable relaxation's runtime of $O(K^3)$ still seems rather expensive. How does this compare to traditional VQ-based methods?
1. Minor weakness in notation: $C$ is both the codebook and the number of channels.

**Questions:**

See weaknesses.

---

> ### Author Response · Authors · 2023-11-15
>
> We thank the reviewer for their comments and feedback. We address the reviewer’s concerns below.
>
> **Regarding weaknesses:**
> 1. SCQ is proposed as a drop-in replacement for VQ within the generative modeling framework. Specifically, our focus in this paper is to introduce SCQ as a relaxation of VQ that effectively addresses several limitations associated with the latter. Through the formulation in Eq. (8) it is clear that VQ is a particular instantiation of SCQ - taking hyperparameter $\lambda$ to infinity exactly recovers the VQ solution. However, by keeping $\lambda$ finite, we obtain a balance between the one-hot VQ solution compatible with downstream generative processes and the improved backpropagation, codebook collapse prevention capabilities of the DCO. Thus we maintain the validity of comparing SCQ with VQ and its variants.
> In this paper the focus was not on any specific downstream generation task, but rather to illustrate SCQ's capabilities as a soft quantization layer that facilitates the training of superior VAE architectures whilst maintaining compatibility with downstream generative models. We anticipate that the significantly enhanced performance of SCQ-embedded VAEs will naturally translate into improved results for downstream generation applications employing these models. In Section 4.2, we demonstrate how SCQGANs consistently outperform VQGANs, especially at smaller latent resolutions. This observation holds valuable implications for downstream generation tasks, where operating on smaller resolutions substantially reduces computational requirements. Looking ahead, our future work aims to integrate SCQ-embedded models with latent generative processes, showcasing the enhanced performance of SCQ in downstream applications.
> 2. Please see the general comments for this concern. We have extended Table 1 (pg. 7) with latency measures of the quantization methods.
> 3. We thank the reviewer for feedback on the overloaded notation. We will correct this.

---

> > ### Comment · Reviewer_Jz15 · 2023-11-17
> > **Response to rebuttal**
> >
> > Thank you for providing quantization latencies.
> >
> > I believe Reviewer d7nB largely shares the same concerns as I do. It still doesn't seem fair to me to compare VQ to SCQ when the former compresses the data, and the latter does not. It is a trivial result for SCQ to give better VAE performance; taken to the extreme, if we completely remove the information bottleneck and replace it with an identity transformation, we could expect even better VAE performance, and with less compute too.
> >
> > Can you elaborate on your response to d7nB "SCQ is readily compatible with latent diffusion processes that operate directly on the encoder embeddings, as well as autoregressive models that require supervision from a categorical distribution over the latent space"? How exactly does this soft "quantized" representation work with downstream generative applications (specifically when $\lambda$ is finite)? Demonstrating this in the paper is critical, because this downstream compatibility is the main thing differentiating SCQ from similarly non-bottlenecked transformations, like identity.

---

> > > ### Author Response · Authors · 2023-11-17
> > >
> > > We thank the reviewer for their additional comments and questions.
> > >
> > > **Comparing VQ and SCQ**
> > >
> > > As mentioned previously, we believe that a comparison between VQ and SCQ is still valid as VQ can be viewed as an instantiation of SCQ with a sufficiently large $\lambda$. Ultimately, we think that the choice for $\lambda$ will be determined based on the downstream application at hand. In the case of generation with an autoregressive (AR) model where VQ is compatible due to its one-hot encoding/discrete latent representation, we will require a sufficiently large $\lambda$ for SCQ where only a “few” codebook vectors are chosen for the soft quantization. The choice for keeping the columns of $P$ as probability distributions in SCQ is so that we can supervise the AR model, e.g. with a cross-entropy loss. For latent diffusion, we believe we can keep a smaller value of $\lambda$ that relaxes the VQ problem more. This is because latent diffusion operates on the encoder embeddings, i.e. the layer preceding the quantization bottleneck.
> > >
> > > The hyperparameter $\lambda$ within the SCQ gives us a choice to mimic VQ or relax the problem to utilizing multiple codebook vectors for soft quantization.
> > >
> > > **Compatibility with downstream generative applications**
> > >
> > > Within generative modeling, the goal for autoencoders with quantization bottlenecks is to extract informative low-dimensional representations within the latent space. This reduces the computational burden on downstream generative tasks. The SCQ method is readily compatible with state-of-the-art latent diffusion models which operate on the encoder embeddings directly (Rombach et al, 2021). Here latent diffusion models are trained on extracted encoder embeddings of the pre-trained VAE. During generation, the latent diffusion models sample new encoder embeddings that are passed through the quantization bottleneck and then the decoder. Relaxing the hard quantization with SCQ doesn’t deter this process of generating encoder embeddings.
> > >
> > > As SCQ inherently has a continuous latent space restricted to the unit simplex, an additional step needs to be taken to extract discrete latents to accommodate generative processing that operate on discrete sequences.
> > >
> > > More specifically, in SCQ the latents can be viewed as probabilities that weight contributions from different codebook vectors in representing encoder embeddings. Larger probabilities suggest that the corresponding codebook vector contributes more significantly to the embedding reconstruction. Also recall that SCQ solves for a set of latent weights in optimization Eq. (8) biased towards the VQ one-hot encoding. Thus, the top-$S$ codebook vectors with largest weights can be considered for reconstruction where $S$ is a hyperparameter to be chosen together with $\lambda$ in optimization Eq. (8) based on the “sparsity” induced. The indices of the top-$S$ vectors can be concatenated to construct a batch of discrete latent images $P_{\textrm{discrete}}\in\mathbb{R}^{N\times S\times \widetilde{H}\times \widetilde{W}}$. Generative autoregressive models can then be trained on these discrete latent images with targets given by the corresponding categorical distribution in columns of $P^\star$.
> > >
> > > We would be happy to make SCQ’s application within downstream generation clearer in the paper and welcome any other questions.

---

### Author Response · Authors · 2023-11-15
**General Comments**

We thank all reviewers for their questions and constructive feedback. Below we address the questions surrounding the missing quantization latency numbers.

**Results on CIFAR-10**
| Method      | MSE ($10^{-3}$)$\downarrow$ | Quant Error$\downarrow$ | Perplexity$\uparrow$ | Avg Quant Time (ms)$\downarrow$ |
| ----------- | ----------- | ----------- | ----------- | ----------- |
| VQVAE  | 41.19 | 70.47 | 6.62 | 4.45 |
| VQVAE + Rep  | 5.49 | $4.13\times 10^{-3}$ | 106.07 | 5.56 |
|VQVAE + Affine + OPT  | 16.92 | $25.34\times 10^{-3}$ | 8.65 | 5.74|
|VQVAE + Rep + Affine + OPT  | 5.41 | $4.81\times 10^{-3}$ | 106.62 | 5.78|
|Gumbel-VQVAE  | 44.5 | $23.29\times 10^{-3}$ | 10.86 | **0.84**|
|RQVAE  | 4.87 | $44.98\times 10^{-3}$ | 20.68 |  12.4|
|SCQVAE  | **1.53** | $\mathbf{0.15}\times \mathbf{10^{-3}}$ | **124.11** | 7.42|

**Results on GTSRB**
| Method      | MSE ($10^{-3}$)$\downarrow$ | Quant Error$\downarrow$ | Perplexity$\uparrow$ | Avg Quant Time (ms)$\downarrow$ |
| ----------- | ----------- | ----------- | ----------- | ----------- |
| VQVAE  | 39.30 | 70.16 | 8.89 | 11.61 |
| VQVAE + Rep  | 3.91 | $1.61\times 10^{-3}$ | 75.51 | 11.93 |
|VQVAE + Affine + OPT  | 11.49 | $13.27\times 10^{-3}$ | 5.94 | 11.71|
|VQVAE + Rep + Affine + OPT  | 4.01 | $1.71\times 10^{-3}$ | 72.76 | 11.70|
|Gumbel-VQVAE  | 56.99 | $47.53\times 10^{-3}$ | 4.51 | **0.85**|
|RQVAE  | 4.96 | $38.29\times 10^{-3}$ | 10.41 | 25.84|
|SCQVAE  | **3.21** | $\mathbf{0.24\times 10^{-3}}$ | **120.55** | 12.93|

We extended Table 1 (pg. 7) in the paper and added numbers for the latency for each of the quantization methods. The table shows a comparison between methods on an image reconstruction task for CIFAR-10 and GTSRB over 5 independent training runs. The same base architecture is used for all methods (see Appendix A for details). All metrics are computed and averaged on the test set.

We measured the average time (in ms) it takes to perform quantization over a single batch of samples (batch size of 128 in considered experiments). We observe that even though SCQ solves a linear system (see Alg. 2) in the forward pass, the quantization time remains competitive with respect to VQ variants. The table demonstrates the benefits of SCQ over VQ variants with respect to reconstruction, quantization errors and codebook usage whilst maintaining a comparable quantization runtime. Note that in the considered experiments on CIFAR-10 and GTSRB we use a codebook size of $K=128$. This is of a similar order of magnitude to the usual codebook sizes used for high resolution images.

**Comments on notation**

We also appreciate the reviewers’ feedback on instances of overloaded notation (specifically in Section 3). We will correct this.